# Revealing the etching process of water-soluble Au$_{25}$ nanoclusters at the molecular level

Yitao Cao[1], Tongyu Liu[2], Tiankai Chen[1], Bihan Zhang[1,3], De-en Jiang [2] & Jianping Xie [1,3✉]

Etching (often considered as decomposition) is one of the key considerations in the synthesis, storage, and application of metal nanoparticles. However, the underlying chemistry of their etching process still remains elusive. Here, we use real-time electrospray ionization mass spectrometry to study the reaction dynamics and size/structure evolution of all the stable intermediates during the etching of water-soluble thiolate-protected gold nanoclusters (Au NCs), which reveal an unusual "recombination" process in the oxidative reaction environment after the initial decomposition process. Interestingly, the sizes of NC species grow larger and their ligand-to-metal ratios become higher during this recombination process, which are distinctly different from that observed in the reductive growth of Au NCs (e.g., lower ligand-to-metal ratios with increasing sizes). The etching chemistry revealed in this study provides molecular-level understandings on how metal nanoparticles transform under the oxidative reaction environment, providing efficient synthetic strategies for new NC species through the etching reactions.

[1] Department of Chemical and Biomolecular Engineering, National University of Singapore, Singapore 117585, Singapore. [2] Department of Chemistry, University of California, Riverside, CA 92521, USA. [3] Joint School of National University of Singapore and Tianjin University, International Campus of Tianjin University, Binhai New City, Fuzhou 735020, China. ✉email: chexiej@nus.edu.sg

Etching was used for decorative purposes in ancient times. In our daily life, metal etching is also one of the most popular chemical reactions. Etching becomes more prominent at the nanoscale-size region due to the high surface energy of metal nanoparticles; therefore, etching is a key consideration in the synthesis, storage, and application of metal nanoparticles[1,2]. For example, the resistance of metal nanoparticles against etching is the prerequisite to their storage and usage[3]. In addition, etching is also indispensable in the ripening process to control the size and shape of metal nanoparticles[4–6]. In particular, etching is the key reaction in understanding the fundamentals of the digestive and Ostwald ripening[7], and plays a vital role in the size-focusing process of metal nanoparticles[8–10]. Therefore, understanding the underlying chemistry of etching will greatly benefit the synthesis, storage, and application of metal nanoparticles.

Recent advances in atomically precise metal nanoclusters (NCs) provide a good platform to understand the underlying etching chemistry of metal nanoparticles[8–12]. Metal NCs are ultrasmall nanoparticles with a core size below 2 nm, and they can be described as $[M_nL_m]^q$, where $n$, $m$, $q$ denote the number of metal atoms, ligands, and net charge, respectively. The molecular formulae of metal NCs can be readily determined by electrospray ionization mass spectrometry (ESI-MS)[13,14], and the ESI-MS can also be used to study the formation processes of metal NCs at the molecular and atomic levels (e.g., using sodium borohydride (NaBH$_4$) or carbon monoxide (CO) as reducing agent)[15–17]. Here, we hypothesize that ESI-MS, especially the real-time ESI-MS, can also be an efficient tool to study the etching processes of metal NCs, providing fundamental understandings on their etching chemistry. Moreover, as etching is a reversed process of growth, the comparison of both processes at molecular level may further reveal the fundamental aspects of the formation and transformation of metal NCs under either reductive or oxidative reaction environment.

Here, we choose water-soluble thiolate-protected Au$_{25}$ NCs (Au$_{25}$SR$_{18}$, SR denotes thiolate ligand) as our model NC. We systematically trace the etching process of water-soluble Au$_{25}$(SR)$_{18}$ in the presence of excess thiol ligands with real-time ESI-MS in 30 days. We discover 20 species formed in total and analyze their variations with time to understand the etching process at molecular precision. As shown in Fig. 1, we reveal that the etching process can be separated into two stages, an initial decomposition process and a subsequent recombination process.

The change of valence electron count ($N^*$, $N^* = m - n - q$, based on the molecular formula $[M_nL_m]^q$)[18] of NCs follows a two-electron-hopping mode. In Stage I, Au$_{25}$(SR)$_{18}$ is first decomposed by thiol radicals to generate NCs with smaller sizes and $N^*$. In Stage II, the recombination process produces isoelectric NCs (i.e., NCs with the same $N^*$) with larger sizes yet also larger ligand-to-metal ratio by isoelectric addition reactions with Au(I)-SR complexes. This process shows prominent differences from the reductive growth process, which leads to NCs of larger sizes but smaller ligand-to-metal ratios. Our findings shed light into how metal nanoparticles transform in oxidative environment at molecular level and provide guidance to synthesize new NC species by etching reactions.

## Results

**Precursor preparation and real-time ESI-MS measurement**. We first synthesized water-soluble [Au$_{25}$(MHA)$_{18}$]$^-$ NCs (MHA = 6-mercaptohexanoic acid, hereafter SR denotes MHA unless otherwise specified) by using the NaBH$_4$-reduction method. To exclude all other by-products and small complexes that could affect the etching process as well as the real-time ESI-MS measurement, we purified the product by ultrafiltration and polyacrylamide gel electrophoresis (PAGE). The etching process was initiated by adding excess thiol ligands (MHA) to the purified Au$_{25}$SR$_{18}$ NCs. We set the ratio of thiol-to-Au in the reaction solution to 2.0 (here the thiol ligands also include those on the original Au$_{25}$SR$_{18}$ NCs). Note that the amount of thiol ligand is excess, which is sufficient enough for the complete etching of Au$_{25}$SR$_{18}$ to generate the smallest Au(I)-SR complexes (i.e., [AuSR$_2$]$^-$). In addition, this value (i.e., 2.0) is equivalent to the feeding ratio of thiol-to-Au in the reductive growth of Au$_{25}$SR$_{18}$ in the previous studies[15,16]. The pH value of the solution is maintained at 9.10 ± 0.20, because this pH can provide the best signal in the ESI-MS measurement, and the pH value remained almost unchanged during the reaction (Supplementary Fig. 1). The reaction solution was exposed to oxygen throughout the reaction process to support the continuous etching process and kept clear throughout the reaction process (no precipitation formed).

Real-time ESI-MS can directly analyze the reaction intermediates in the reaction solution without further purification, which enables the monitoring of evolution of the intermediate species throughout the etching process[14] of [Au$_{25}$(MHA)$_{18}$]$^-$. As shown in Fig. 2a, the mass spectra of the reaction solution were recorded over a period of 30 days. At the beginning, the mass spectra only showed signals from pure Au$_{25}$SR$_{18}$, and the zoom-in spectrum revealed that after purification, most of them are oxidized to the species with a charge state of 0 ([Au$_{25}$(MHA)$_{18}$]$^0$, Supplementary Fig. 6)[19]. Thus, $N^*$ of the NC precursor (Au$_{25}$SR$_{18}$) was in a mixed state of 8 and 7. The $N^*$ of the intermediate species decreased in the continuous oxidative etching process of Au NCs. During the 30-day etching process, we have captured 20 intermediates (see Supplementary Figs. 7–23 for detailed analysis), all of which feature an even number $N^*$, ranging from 6 to 0 (Fig. 2b). These data also suggest that the etching process of Au NCs also follows a two-electron-hopping mode (i.e., $N^*$ decreases by 2), similar to the two-electron-hopping trend observed in the reductive growth of Au NCs ($N^*$ increases by 2)[15–17]. The normalized ESI-MS spectral intensities of all the identified species over the 30 days etching were analyzed in Fig. 2c. Three batches of experiments were conducted to ensure the reproducibility of the etching process. In addition, PAGE was used to separate the product species, and it was confirmed that no in-situ electrochemical reactions occurred under the electrospray conditions of ESI-MS (Supplementary Fig. 24).

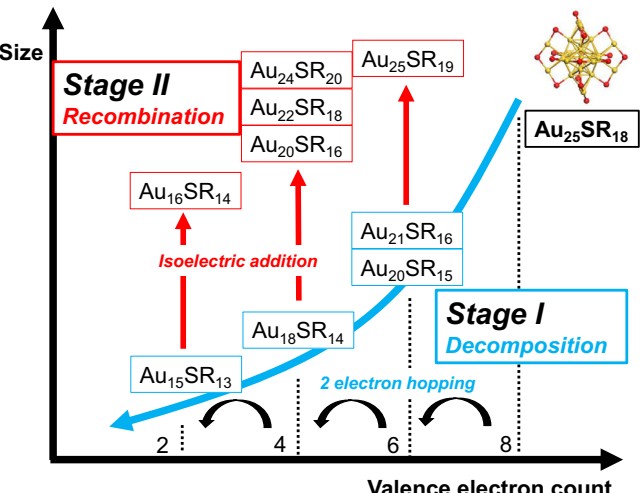

**Fig. 1 Schematic illustration of the etching process.** Two stages dominated by different reaction processes are shown. Stage I: decomposition and Stage II: recombination.

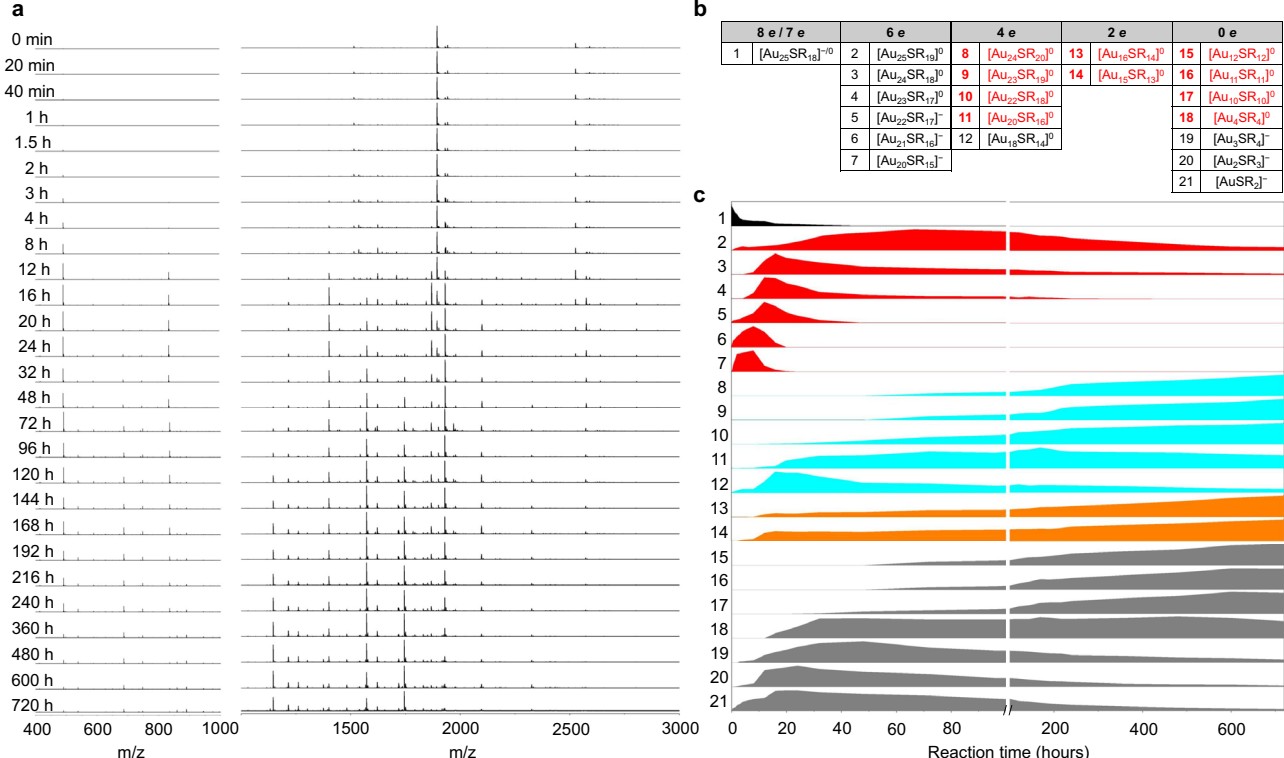

**Fig. 2 Real-time ESI-MS measurement of the etching process. a** Time-course ESI mass spectra of the reaction solution of $[Au_{25}(MHA)_{18}]^-$ in the presence of excess MHA ligands. Enlarged view of different parts are also shown in Supplementary Figs. 2–5 for a better overview of the ESI mass spectra in 30 days. **b** Molecular formulae of all 21 species identified in the entire etching process (30 days). These species were classified by their valence electron counts ($N^*$). Species in red are those survived as the main products after 30 days of etching. **c** Normalized ESI-MS spectral intensity profiles of the 21 species throughout the etching process (30 days). Species with different $N^*$ were marked with different colors. Black: $N^* = 7$ and 8; red: $N^* = 6$; cyan: $N^* = 4$; orange: $N^* = 2$; gray: $N^* = 0$.

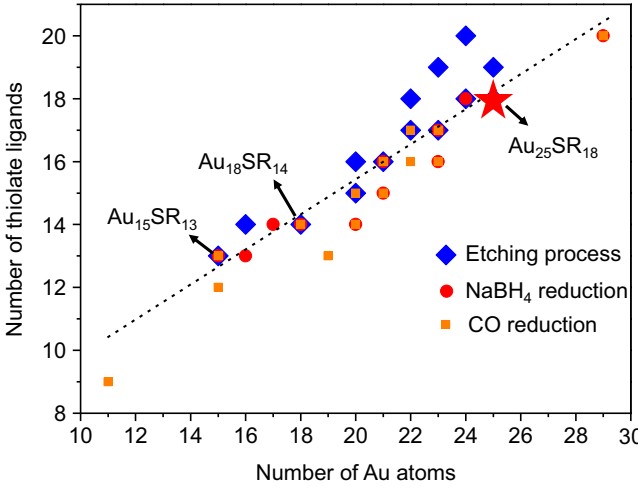

**Fig. 3 NC species detected in different reaction process.** Comparison of the NC species detected during the etching (blue square) and reductive growth (using $NaBH_4$ (red circle) or CO (orange square) as reducing agent) of water-soluble $Au_{25}SR_{18}$. The black dashed line is a guideline for comparing the relative positions of species.

## Comparison of the intermediates in etching and reduction process

Next, we may ask whether etching of $Au_{25}SR_{18}$ is the reverse process of its growth. In order to explore the similarities and differences between the etching and growth processes of $Au_{25}SR_{18}$, we plotted all NC species ($N^* > 0$) that have been identified during the etching process (Fig. 3), together with the intermediate species ($N^* > 0$) that have been identified previously in the reductive growth of $Au_{25}SR_{18}$, by using CO or $NaBH_4$ as reducing agent (please refer to Supplementary Tables 1 and 2 for the detailed molecular formulae of these NC species)[15,16]. The dashed line is a guideline to compare the relative position of these species. Some well-known magic-sized NC species are located along the dashed line, including $Au_{15}SR_{13}$, $Au_{18}SR_{14}$, and $Au_{25}SR_{18}$. By carefully comparing the relative positions of these species, one can see that the etching process is not a reversed process of the NC growth, but follows different reaction pathways. The intermediate species generated in the reductive environment in the growth process of Au NCs are below the dashed line, which indicates that these species have relatively lower ligand-to-metal ratios. In contrast, in the etching process, although some intermediate species near the dashed line are shared with the growth process, a number of intermediate species are above this line, featuring higher ligand-to-metal ratios, and most of these species were survived after 30 days of etching (Fig. 2b, c).

## Two-stage reaction process

In order to better illustrate the entire etching process and find out how the species with high ligand-to-metal ratios were formed, in Fig. 4, we plotted the full width at half maximum (FWHM) of the normalized ESI-MS spectral intensity profiles of all the species in Fig. 2c. The data were presented by the projective segment of the reaction time during the etching process, from which we can obtain the formation and consumption process of each species and their relative lifetime during the etching process.

The entire etching process can be roughly divided into two stages: Stage I, decomposition; and Stage II, recombination. Stage

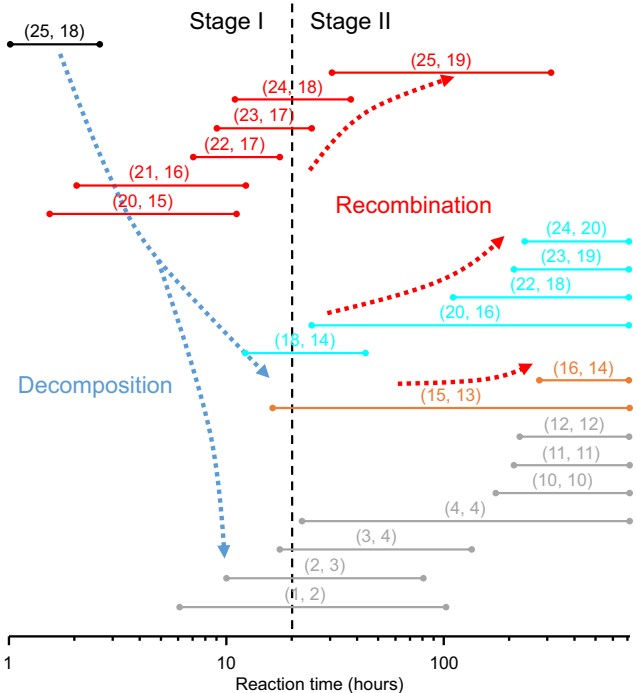

**Fig. 4 Two-stage reaction process.** Segments representing the full width at half maximum (FWHM) of the normalized ESI-MS spectral intensity profiles in Fig. 2c. ($m$, $n$) denotes species with the molecular formula $Au_mSR_n$. Species were classified by $N^*$ and marked with different colors. Black: $N^* = 7$ and 8; red: $N^* = 6$; cyan: $N^* = 4$; orange: $N^* = 2$; gray: $N^* = 0$. Species with the same $N^*$ were arranged according to their relative size. The entire reaction process can be well divided into Stage I (decomposition process) and Stage II (recombination process).

I is related to the rapid consumption of $Au_{25}SR_{18}$ in the first 20 h, which produces a series of smaller species, such as $[Au_{20}SR_{15}]^-$, $[Au_{21}SR_{16}]^-$, $[Au_{18}SR_{14}]^0$, $[Au_{15}SR_{13}]^0$, and short Au(I)-SR complexes ($[AuSR_2]^-$ and $[Au_2SR_3]^-$). We have successfully captured the thiol radicals in our reaction system (Supplementary Fig. 25). Thiol radicals can be generated by the reaction between $O_2$ and thiol ligands, and the reactions mediated by thiol radicals should drive the decomposition process in Stage I[20]. Considering the sequence of formation of these smaller species, the initial reactions should be the etching of $Au_{25}SR_{18}$ by thiol radicals to generate $[Au_{20}SR_{15}]^-$ and $[Au_{21}SR_{16}]^-$, as well as short Au(I)-SR complexes (Fig. 4, blue dashed line). Both $[Au_{20}SR_{15}]^-$ and $[Au_{21}SR_{16}]^-$ have a relatively short lifetime, and would be further etched by thiol radicals to generate $[Au_{18}SR_{16}]^0$ and $[Au_{15}SR_{13}]^0$. The detailed reactions are presented in Supplementary Fig. 26. Interestingly, all species ($N^* > 0$) formed at this stage can also be detected in the growth process of $[Au_{25}SR_{18}]^-$ (see Supplementary Table 3 for detailed comparison). It appears that at least in Stage I, the etching process shares some similarities with the reverse growth process of Au NCs.

In order to further verify the role of thiol ligands and $O_2$ in the decomposition process, we performed several control experiments in an oxidative environment without free thiol ligands and in a $CO/N_2$ atmosphere with free thiol ligands. As shown in Supplementary Figs. 27 and 28, $Au_{25}SR_{18}$ shows much higher stability without free thiol ligands, and its decomposition process is quenched by saturating the reaction system with CO or $N_2$. Therefore, the control experiments suggest that free thiol ligands and $O_2$ are both essential in the etching process of $Au_{25}SR_{18}$.

**Recombination process in etching reaction and mechanism investigation.** In Stage I, several other species with $N^* = 6$ are also formed, such as $[Au_{22}SR_{17}]^-$, $[Au_{23}SR_{17}]^0$, and $[Au_{24}SR_{18}]^0$. However, there is a clear time mismatch between the formation of these 6-electron species and the consumption of $Au_{25}SR_{18}$. On the contrary, in these 6-electron species, the formation sequence appears stepwise from the smaller species to the larger ones (red dashed line). This observation clearly suggests a recombination process between the early-formed species (e.g., $[Au_{20}SR_{15}]^-$ and $[Au_{21}SR_{16}]^-$) and the short Au(I)-SR complexes, most likely through an isoelectric addition reaction (see Supplementary Fig. 29 for the proposed reactions). This recombination process becomes more dominant in Stage II, which produces a series of NC species with the same $N^*$ (4 and 2) but of larger sizes (Fig. 4, red dashed lines in Stage II). These species should also be produced by recombination of smaller isoelectric species with short Au(I)-SR complexes, evidenced by their stepwise formation sequence and simultaneous consumption of small species (e.g., $[Au_{18}SR_{14}]^0$) and short Au(I)-SR complexes (e.g., $[AuSR_2]^-$, $[Au_2SR_3]^-$, and $[Au_3SR_4]^-$). It should be noted that the change of dominant reaction routes in different stages was not due to the consumption of free thiol ligands, because excess thiol ligands were introduced and thiol radicals can still be captured in Stage II (Supplementary Fig. 25).

The recombination process involves the combination between Au NCs and Au(I)-SR complexes to produce NC species with larger size and relatively higher ligand-to-metal ratio, which has never been observed in the reductive growth process of Au NCs (Supplementary Table 3)[15,16]. This recombination process is the main feature of the etching of Au NCs and is distinctly different from the reverse growth process of Au NCs. In order to gain more insights into the recombination process of etching, we first confirmed the reaction mechanism we proposed in the 4-electron species by using a control experiment with pure $[Au_{18}SR_{14}]^0$. In the presence of Au(I)-SR complexes, $[Au_{18}SR_{14}]^0$ can be converted to larger $[Au_{20}SR_{16}]^0$ and $[Au_{22}SR_{18}]^0$. As shown in Fig. 5a, after 24 h of reaction, the sample with Au(I)-SR complexes showed the prominent formation of $[Au_{20}SR_{16}]^0$ and $[Au_{22}SR_{18}]^0$. In contrast, in the control experiment without Au (I)-SR complexes, the signal corresponding to these larger species only increased slightly. The slight increase in these larger species may be due to the slow decomposition of the original $[Au_{18}SR_{14}]^0$ as we did not remove oxygen in the reaction. The driving force for the recombination process of 4-electron species was determined by DFT calculation (Fig. 5b), which indicates that the isoelectric addition reaction through stepwise reaction with $[Au_4SR_4]^0$ is energetically preferred. The calculated energy from $[Au_{18}SR_{14}]^0$ to $[Au_{20}SR_{16}]^0$ is $-22.3$ kcal/mol, and from $[Au_{18}SR_{14}]^0$ to $[Au_{22}SR_{18}]^0$ is $-21.0$ kcal/mol. It should be noted that the reactant $[Au_4SR_4]^0$ involved in the recombination process can be captured throughout the Stage II, and this species was formed by consuming smaller Au(I)-SR complexes, such as $[AuSR_2]^-$, $[Au_2SR_3]^-$, and $[Au_3SR_4]^-$, according to their relative formation sequence and consumption process (Fig. 4).

There is also a clear trend that the larger species formed during the recombination process have a longer lifetime than the smaller isoelectric species, which indicates that they are more stable in (i.e., they are more resistance to) an oxidative environment. The high stability of NCs may originate from the long-chain motifs on the Au(0) core, which become dominant on the NC surface with the addition of Au(I)-SR complexes[21–25]. The variations in surface motifs in the recombination process were verified by tandem mass spectrometry (MS/MS), which is sensitive to the ligand shell of metal NCs[26,27]. As shown in Fig. 5c, the 1st-generation fragmentation pathway of a series of 4-electron species clearly showed that the length of the fragments

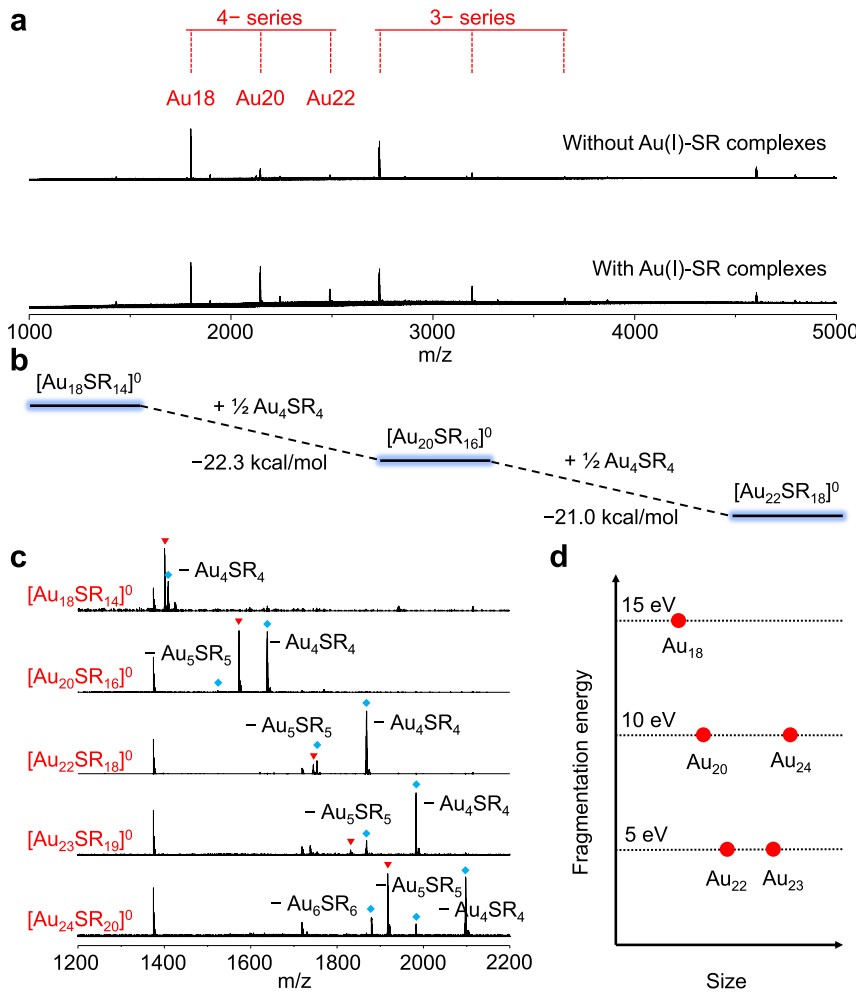

**Fig. 5 Isoelectric addition reactions and MS/MS spectra of 4-electron species. a** ESI mass spectra of the reaction solution of pure $[Au_{18}SR_{14}]^0$ with or without the addition of Au(I)-SR complexes. **b** Calculated reaction energy of isoelectric addition reaction involving 4-electron species $[Au_{18}SR_{14}]^0$. **c** MS/MS spectra and corresponding fragmentation pathways of 4-electron species when the collision energy is 15 eV. The peaks corresponding to the parent species and the 1st-generation fragment species are marked with red ▼ and blue ◆, respectively. More intense fragmentations were observed as the sizes of the NC species increased. **d** Changes in the on-set fragmentation energy of 4-electron species. See Supplementary Figs. 30–34 for more detailed MS/MS spectra and analysis.

increases with increasing size: from $[Au_4SR_4]$ in $[Au_{18}SR_{14}]^0$, to $[Au_5SR_5]$ in the medium-sized species, and finally to $[Au_6SR_6]$ in the largest $[Au_{24}SR_{20}]^0$. The lengthened fragments in MS/MS spectra are in good accordance with the dominant lengthened motifs on their surface as inferred from the well-resolved structures of $[Au_{18}SR_{14}]^0$, $[Au_{20}SR_{16}]^0$, $[Au_{22}SR_{18}]^0$, and $[Au_{24}SR_{20}]^0$ [21–25]. There is also a trend that larger species will fragment more intensely under the same collision energy (Fig. 5c), while their on-set fragmentation energy becomes lower (Fig. 5d). It should be noted that the on-set fragmentation energy of $[Au_{24}SR_{20}]^0$ (10 eV) is relatively higher than that of $[Au_{22}SR_{18}]^0$ and $[Au_{23}SR_{19}]^0$ (5 eV), which may be due to the unique interlocked surface motifs of $[Au_{24}SR_{20}]^0$ [25]. The inverse correlation between the on-set fragmentation energy and NC size is also closely related to the entangled motifs on the Au(0) core, which becomes more rigid as the surface motifs lengthen [25,28], and more easily dissociate the fragments to release the surface tension. We believe that the lengthened motifs and rigidified ligand shell will reduce the accessibility of thiol radicals to the Au (0) core, which will slow down (if not completely quench) the oxidative etching reaction and thus enrich these species in oxidative environment.

## Discussion

The recombination process between Au NCs and Au(I)-thiolate complexes well explains the phenomenon observed in previous studies: $[Au_{22}SR_{18}]^0$ and $[Au_{24}SR_{20}]^0$ were successfully synthesized in an oxidative environment [25,28]. Furthermore, as the rigidified ligand shell has close relationship with the increased fluorescence in metal NCs [29], the recombination process also provides possible explanation to the formation of fluorescent metal NCs in oxidative etching reactions [30–34]. It is also worth noting that another kind of recombination process was observed during the ligand-exchange-induced size/structure transformation process [35–37], usually by the fusion of the metallic cores of two separate clusters, which is different from our observations in the etching reaction.

Another feature of Stage II is that in the presence of oxygen and excess free thiol ligands, the oxidative etching process is continuously but greatly slowed down, which helps explain some observations in the etching process of Au NCs, including the consumption of $[Au_{25}SR_{19}]^0$ and the formation of long-chain Au (I)-SR complexes ($[Au_{10}SR_{10}]^0$, $[Au_{11}SR_{11}]^0$, and $[Au_{12}SR_{12}]^0$). For example, the oxidative etching process of $[Au_{25}SR_{19}]^0$ took more than 500 h, which is much longer than that of $Au_{25}SR_{18}$

(~20 h). These data indicate that the lengthened Au(I)-SR motifs on the $[Au_{25}SR_{19}]^0$ surface can greatly reduce the reaction dynamics during the etching process. In Stage II, long-chain Au (I)-SR complexes ($[Au_{10}SR_{10}]^0$, $[Au_{11}SR_{11}]^0$, and $[Au_{12}SR_{12}]^0$) were also formed at a very low rate. In principle, there are two possible pathways to generate these long-chain Au(I)-SR complexes. One is due to the recombination of small Au(I)-SR complexes (e.g., $[AuSR_2]^-$), and the other is due to the oxidative etching of larger NC species. However, we propose that the long-chain Au(I)-SR complexes should only be derived from the continuous oxidative etching of the larger NC species at a very low rate. This is because there is an obvious time mismatch between the consumption of small Au(I)-SR complexes and the formation of these long-chain Au(I)-SR complexes. In addition, we did not capture any intermediate species whose size is between $[Au_4SR_4]^0$ and $[Au_{10}SR_{10}]^0$. Therefore, the largest Au(I)-SR complexes produced by recombination of the small Au(I)-SR complexes should be $[Au_4SR_4]^0$. Hence, we propose some typical oxidative etching reactions in Stage II in Supplementary Fig. 35.

In summary, we monitored the etching process of water-soluble $Au_{25}SR_{18}$ NCs in the presence of excess thiol ligands by real-time ESI-MS within 30 days. Twenty-one species have been identified, and their time-course formation and consumption trends have been used to construct the etching process of Au NCs. We found that although the decrease in valence electron counts of these species still follows a two-electron-hopping mode during the entire etching reaction, the etching process and the reverse growth process show distinct differences, and the etching process can produce NC species that have never been obtained in the reductive growth reaction. The entire etching process can be divided into two stages: Stage I, decomposition; and Stage II, recombination. During the initial decomposition process, $Au_{25}SR_{18}$ were etched by thiol radicals, forming small species and short Au(I)-SR complexes within 20 h. In the subsequent recombination process, the small NCs will react with the short Au (I)-SR complexes through isoelectric addition reactions to generate a larger species with a higher ligand-to-metal ratio. The isoelectric addition reaction was confirmed by control experiments and DFT calculations using the 4-electron species as a model. MS/MS proves that the lengthened Au(I)-SR motifs on the Au(0) core are responsible for increasing the stability of larger isoelectric species in an oxidative environment. Our research will promote the molecular-level understanding of the formation and transformation process of metal NCs, and help discover new NC species and their corresponding synthetic methods in oxidative environment.

## Methods

**Materials**. Hydrogen tetrachloroaurate(III) hydrate ($HAuCl_4·3H_2O$) was purchased from Alfa Aesar; 6-mercaptohexanoic acid (MHA), sodium hydroxide (NaOH), and sodium borohydride ($NaBH_4$) were purchased from Sigma-Aldrich. All chemicals were used without further purification. Ultrapure water (18.2 MΩ·cm) was used in all the experiments. All glassware was washed with aqua regia before use.

**Synthesis of pure Au$_{25}$(MHA)$_{18}$ NCs**. $Au_{25}(MHA)_{18}$ NCs were synthesized by a reported $NaBH_4$-reduction method[38], followed by a native PAGE separation. In a typical synthesis, 4 mL of aqueous solution of 5 mM MHA and 0.2 mL of aqueous solution of 50 mM $HAuCl_4$ were added into 5 mL of ultrapure water. After 5 min, the pH of the solution was brought up by adding 0.15 mL of aqueous solution of 1 M NaOH. After 10 min, 0.20 mL of aqueous solution of $NaBH_4$ (prepared by dissolving 43 mg of $NaBH_4$ in 10 mL of 0.2 M NaOH) was introduced into the solution to initiate the synthesis of Au NCs. The reaction was allowed to proceed for 3 h under gentle stirring (500 rpm) at room temperature. In the subsequent native PAGE experiments, Bio-Rad Mini-PROTEAN® Tetra Cell or PROTEAN® II xi Cell system was used. Stacking and resolving gels were prepared by 4 and 30 wt% acrylamide monomers (1:19), respectively. The Au NC solutions were first concentrated by ultrafiltration and then mixed with glycerol to prepare the sample solutions (~10 mM based on Au atoms, 50 vol% glycerol); 2 mL of the sample

solution was loaded into the well. PAGE was conducted with a constant voltage of 180 V at 4 °C for 24 h. After that, the bands were cut, crushed, and incubated in ultrapure water to obtain solutions of pure Au NCs for further use. The NC concentration was determined by inductively coupled plasma mass spectrometry (ICP-MS).

**Etching of Au$_{25}$(MHA)$_{18}$ NCs in the presence of excess ligands**. Here, 256 μL of aqueous solution of MHA (5 mM) was introduced into 5 mL of aqueous solution of purified $Au_{25}(MHA)_{18}$ NCs (0.2 mM Au atom basis). The pH was kept at 9.10 ± 0.20. Glass containers were used in the experiments. The container was saturated with air after adding the thiol ligands to initiate the etching process, and it was kept airtight. The solution for testing was drawn out by a syringe to avoid possible loss of solvent. The temperature was kept at 25 °C. The container was also kept in dark to avoid possible influence of light. Three batches of experiments were conducted to ensure the reproducibility of the etching process. The reaction solution was directly purged into ESI-MS instrument without further purification, enabling a real-time ESI-MS monitoring of the etching process.

**Isoelectric addition reactions involving 4-electron species**. Using the same method as $Au_{25}(MHA)_{18}$, $[Au_{18}SR_{14}]^0$ NCs were separated by PAGE from the solution in the etching process. Au(I)-SR complexes were prepared by mixing 4 mL of aqueous solution of MHA (5 mM) and 0.2 mL of aqueous solution of $HAuCl_4$ (50 mM) in 5 mL of water. pH was adjusted to 12.0 and the solution was stirred for 30 min before further use. The reaction solution was prepared by mixing pure $[Au_{18}SR_{14}]^0$ solution with Au(I)-SR complexes while keeping the final concentration of $[Au_{18}SR_{14}]^0$ at 50 μM (Au atom basis) and Au(I)-SR complexes at 10 μM (Au atom basis). The reaction solution was kept at room temperature for 24 h and characterized by ESI-MS without further purification.

**Characterizations**. Solution pH was measured by Mettler-Toledo FE 20 pH-meter. UV-Vis absorption spectra were recorded by Shimadzu UV-1800 spectrometer. ESI mass spectra were captured by a Bruker microTOF-Q system under negative ion mode. ESI-MS testing parameters: source temperature 120 °C, dry gas flow rate 8 L per min, nebulizer pressure 3 bar, capillary voltage 3.5 kV, and sample injection rate 3 μL per min. In ESI-MS test, the solution was directly injected without further purification.

## Data availability

All relevant data are available from the corresponding authors on request.

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

## Acknowledgements

We acknowledge the financial support from the Ministry of Education, Singapore, Academic Research Grant R-279-000-580-112, R-279-000-538-114 and R-279-000-634-114. DFT computation was supported by the U.S. Department of Energy, Office of Science, Office of Basic Energy Sciences, Chemical Sciences, Geosciences, and Biosciences Division.

## Author contributions

J.X. and Y.C. conceived the idea and designed the experiments. J.X. supervised the project. Y.C. carried out the experiments and characterizations. T.L. and D.J performed the DFT calculations. Y.C. and J.X. wrote the manuscript. T.C. and B.Z. discussed the results and commented on the manuscript.

## Competing interests

The authors declare no competing interests.
