## [Peer Review File. · Nature Communications]

Reviewers' Comments:

Reviewer #1:

Remarks to the Author:

Xie et al. report their ESI-MS measurements on the real time composition change of [Au₂₅(MHA)₁₈]⁻. The etching process starting with ligand to metal ratio at 2:1 gives interesting results that the process includes two stages: decomposition and recombination. The data coming from 30 days' monitoring are informative, and this work will be helpful for understanding of the formation and transformation process of nanoclusters. Two important points need the authors' concern:

1. The stability of [Au₂₅(MHA)₁₈]⁻ (no additional ligands are added) should be also monitored with ESI-MS, in order to make comparison between etching and non-etching cases.
2. I understand this is a time consuming work, but it is worth checking the difference under reductive and oxidative conditions. Will larger clusters be observed under reductive atmosphere? How about in the absence of oxygen.

Reviewer #2:

Remarks to the Author:

Cao et al. report the in-depth analysis of the etching mechanism of water-soluble Au₂₅ nanoclusters. Specifically, ESI-MS was primarily used as the experimental technique to map out the etching mechanism consisting of two stages: (1) Decomposition, followed by (2) Recombination. Since this work attempts to provide detailed insights into understanding the ubiquitous process of etching in nanomaterials, this paper will appeal to a broad community but the following concerns must be addressed before publication of this manuscript.

1) Since ESI-MS is the primary experimental technique used in the study, can the authors confirm that in-situ electrochemical reactions are not occurring under electrospray conditions? This possibility is especially relevant since the reaction solutions are directly injected into the ESI-MS without purification. Have the authors tried some sort of purification, e.g., fractional precipitation of different species appearing at time points > 16 h? Such a control will make their results unambiguous.

2) This manuscript does not comprehensively discuss previous work on etching of nanomaterials. E.g., there has been a previous study (Chem. Mater. 2016, 28, 4, 1022–1025) by Li et al. on the etching mechanism of an organic soluble Au₂₅ nanocluster by phosphine ligands. In this case, the mechanism consisted of partial decomposition of parent cluster to a smaller cluster followed by recombination leading to the formation of a new bigger cluster. Hence, "unexpected recombination" (line 22) is not really true!

3) Can the authors comment on the generality of the elucidated mechanism? What factors play the most important role- temperature, specific ligands, etc?

4) References should be placed after the comma/period.

Reviewer #3:

Remarks to the Author:

see attached

Review of "Revealing the etching process of water-soluble Au₂₅ nanoclusters at the molecular level" by Xie et al.

This publication reports an ESI- mass spectroscopic study of aqueous solutions of Au₂₅(MHA)₁₈ with excess added MHA at pH=9. In addition to the temporal development of high resolution mass spectra serially acquired over an unusually long time period ranging up to 700 hours after mixing, the authors report collision induced dissociation measurements and density functional theory calculations of selected species generated during the corresponding "etching" process. This leads sequentially to smaller clusters. Surprisingly there are also indications of addition reactions which set in at intermediate times.

I have several problems with this manuscript which lead me to recommend rejection:

(i) While pH is apparently held constant over more than 700 hours nothing is said about possibly changing solution concentration. What about solvent evaporation/condensation or adsorption of reactants on the container walls? What container material? To what extent is the kinetic dependence observed in Figure 2 (which by the way is almost unreadable) reproducible? How does it depend on temperature (was the solution thermostatted and held under constant humidity)?

(ii) MHA complexes were studied by negative ion mass spectroscopy relying on deprotonation of the MHA ligands for ionization. For example, primarily the tetraanionic species $[\text{Au}_{25}(\text{MHA})_{18} - 4\text{H}^+]^{4-}$ was in fact initially probed - rather than "Au₂₅(MHA)₁₈" or "Au₂₅(MHA)₁₈⁻" as discussed in the main text. This tetraanion is then etched to smaller trianions and dianions rather than to the neutral or monoanionic species discussed in the main text. Presumably the authors have chosen to neglect this in the main text to make things clearer for the reader. The opposite is in fact the case requiring this reader to go back and forth between supplement and main text to understand. Apart from terminology there is another possibly severer buried issue here: J.Phys.Chem. A 2020, 124, 5840–5848 has suggested that there are "charge isomers" in related MBA terminated Au₂₅ clusters, i.e. the cores of nominally equi m/z species can have a range of different charges. The authors should also report changes to other charge states bearing the same gold/ligand compositions (e.g. the parent tetraanion also comes as a (weaker) trianion and pentaanion).

(iii) I appreciate that this is probably one of the first studies in this field to observe long-term etching phenomena by mass spectroscopy. But the authors devote a lot of additional space to making mechanistic statements which in my view are not sufficiently supported by the data. If they want to discuss mechanism they should provide a quantitative kinetic model and compare it to their observations.

Replies to reviewers' comments and descriptions of revisions made

Comments by Reviewer #1:

General comments. Xie et al. report their ESI-MS measurements on the real time composition change of $[Au_{25}(MHA)_{18}]^-$. The etching process starting with ligand to metal ratio at 2:1 gives interesting results that the process includes two stages: decomposition and recombination. The data coming from 30 days' monitoring are informative, and this work will be helpful for understanding of the formation and transformation process of nanoclusters. Two important point need the authors' concern:

Reply: We really appreciate the reviewer's constructive comments and suggestions, which have greatly improved the quality of our paper. Thank you.

Comment 1. The stability of $[Au_{25}(MHA)_{18}]^-$ (no additional ligands are added) should be also monitored with ESI-MS, in order to make comparison between etching and non-etching cases.

Reply: Thank you for the constructive suggestion. This point should be an important supplementary to our study on the etching process. As suggested, we studied the stability of $Au_{25}MHA_{18}$ nanoclusters without the addition of MHA ligands under aerobic conditions. According to the ESI-MS data in Figure R1, $Au_{25}MHA_{18}$ nanoclusters show much higher stability in the absence of excess MHA ligands. Only slight decomposition was observed within 15 days.

Figure R1. ESI mass spectra of $Au_{25}MHA_{18}$ nanoclusters without adding excess MHA ligands.

Revisions

Page 10, Lines 9-14

“In order to further verify the role of thiol ligands and O_2 in the decomposition process, we performed several control experiments in an oxidative environment without free thiol ligands and

in a CO/N₂ atmosphere with free thiol ligands. As shown in Supplementary Fig. 26 and 27, Au₂₅SR₁₈ shows much higher stability without free thiol ligands, and its decomposition process is quenched by saturating the reaction system with CO or N₂. Therefore, the control experiments suggest that free thiol ligands and O₂ are both essential in the etching process of Au₂₅SR₁₈.”

Supplementary Fig. 26

Figure R1 was included as Supplementary Fig. 26 in the revised Supplementary Information.

Comment 2. *I understand this is a time consuming work, but it is worth checking the difference under reductive and oxidative conditions. Will larger clusters be observed under reductive atmosphere? How about in the absence of oxygen.*

Reply: A very good point. Thank you. As suggested, we conducted control experiments by saturating the reaction system with CO and N₂. As shown in Figure R2, in the presence of excess MHA ligands, Au₂₅MHA₁₈ nanoclusters remained stable for 2 days. Please note that during this reaction time, Au₂₅MHA₁₈ nanoclusters should be completely decomposed under aerobic conditions (Figure 2). We also cannot observe the formation of any species larger than Au₂₅MHA₁₈ in the two reactions.

Figure R2. ESI mass spectra of reaction solution saturated by (a) CO and (b) N₂ in 48 hours.

Revisions

Page 10, Lines 9-14

“In order to further verify the role of thiol ligands and O₂ in the decomposition process, we performed several control experiments in an oxidative environment without free thiol ligands and in a CO/N₂ atmosphere with free thiol ligands. As shown in Supplementary Fig. 26 and 27,

Au₂₅SR₁₈ shows much higher stability without free thiol ligands, and its decomposition process is quenched by saturating the reaction system with CO or N₂. Therefore, the control experiments suggest that free thiol ligands and O₂ are both essential in the etching process of Au₂₅SR₁₈.”

Supplementary Fig. 27

Figure R2 was included as Supplementary Fig. 27 in the revised Supplementary Information.

Comments by Reviewer #2:

General comments. *Cao et al. report the in-depth analysis of the etching mechanism of water-soluble Au₂₅ nanoclusters. Specifically, ESI-MS was primarily used as the experimental technique to map out the etching mechanism consisting of two stages: (1) Decomposition, followed by (2) Recombination. Since this work attempts to provide detailed insights into understanding the ubiquitous process of etching in nanomaterials, this paper will appeal to a broad community but the following concerns must be addressed before publication of this manuscript.*

Reply: We are thankful for the reviewer's constructive comments/suggestions and have tried our best to address all the issues raised by the reviewer.

Comment 1. *Since ESI-MS is the primary experimental technique used in the study, can the authors confirm that in-situ electrochemical reactions are not occurring under electrospray conditions? This possibility is especially relevant since the reaction solutions are directly injected into the ESI-MS without purification. Have the authors tried some sort of purification, e.g., fractional precipitation of different species appearing at time points > 16 h? Such a control will make their results unambiguous.*

Reply: Thank you for the insightful suggestion. We do agree with the reviewer that a control experiment that can eliminate the influence of in-situ electrochemical reactions in ESI-MS will make our results unambiguous. We tried to precipitate our species at different pH values, but failed because the amount of nanoclusters used in our etching reaction was not sufficient for a fractional precipitation process. Therefore, we tried to use polyacrylamide gel electrophoresis (PAGE) to analyze the etching products at a reaction time of 2 days. As shown in Figure R3, we were able to cut three bands from the gel and analyze their molecular formula by ESI-MS. The ESI mass spectra show signals of Au₁₈MHA₁₄, Au₂₂MHA₁₈ and Au₂₅MHA₁₉ for each band, stepwise from small to large. These observations support our ESI-MS data. We think it is reasonable to say that these species exist during the etching process of Au₂₅MHA₁₈ nanoclusters, and no in-situ electrochemical reactions occurred in the ESI-MS test. It is worth noting that the resolution of PAGE is not sufficient to separate all species of similar size (band merging is severe), and some species may not be abundant enough to show clear bands. We also cannot ensure the stability of all species during the long electrophoresis process (usually 24 h). Therefore, we did not use PAGE for fulltime screening of all intermediate species.

Figure R3. PAGE separation of the etching product and the ESI mass spectra of the species from the corresponding bands.

Revisions

Page 6, Lines 14-16

“In addition, PAGE was used to separate the product species, and it was confirmed that no in-situ electrochemical reactions occurred under the electrospray conditions of ESI-MS (Supplementary Fig. 24).”

Supplementary Fig. 24

Figure R3 has been included as Supplementary Fig. 24 in the revised Supplementary Information.

Comment 2. *This manuscript does not comprehensively discuss previous work on etching of nanomaterials. E.g. there has been a previous study (Chem. Mater. 2016, 28, 4, 1022–1025) by Li et al. on the etching mechanism of an organic soluble Au₂₅ nanocluster by phosphine ligands. In this case, the mechanism consisted of partial decomposition of parent cluster to a smaller cluster followed by recombination leading to the formation of a new bigger cluster. Hence, “unexpected recombination” (line 22) is not really true!*

Reply: Thanks for your insightful suggestions, and we apologize for the confusion. We would like to

change the phrase to ‘unusual recombination’. In addition, since this comment is related to the novelty of our work, we hope to further clarify the differences between the mentioned paper and our work here.

Indeed, in the above-mentioned paper, Li et al. observed that the recombination of two metallic Au₁₃ cores led to the formation of a larger cluster. The widely accepted definition of this reported reaction is a ligand-exchange-induced size/structure transformation, which has been extensively studied by Jin’s group (e.g., *J. Phys. Chem. Lett.* **6**, 2976 (2015); *Chem. Rev.* **116**, 10346 (2016)). In similar reactions reported, this is not the only case that occurs through the fusion/recombination of the metallic cores of two separated metal nanoclusters. For example, in the ligand-exchange-induced transformation from Ag₂₅ to Ag₄₄, a dimerization of two Ag₂₅ nanoclusters has been observed (*Chem. Mater.* **28**, 3292 (2016)). The ligand-exchange-induced size/structure transformation may be initiated by the etching of parent clusters, but whether it is appropriate to define it as an etching reaction is still ambiguous.

In our case, the ‘recombination’ is realized during the etching process of Au₂₅ nanoclusters on a single cluster (not by fusing two metallic cores), which produces cluster species with larger size and higher ligand-to-metal ratio by combining Au(I)-SR complexes. Considering the common sense of an etching process to be a purely top-down process (e.g., nanoparticles become smaller and smaller during the etching process), we believe our findings are novel.

Accordingly, in the revised manuscript, we also provide additional discussion on the recombination process in the ligand-exchange-induced size/structure transformation, and better clarify our definition of recombination to avoid possible misunderstandings. Thank you.

Revisions:

Page 2, Lines 6-7

“an unusual “recombination” process”

Page 11, Lines 8-9

“The recombination process involves the combination between Au NCs with Au(I)-SR complexes to produce NC species with larger size and relatively higher ligand-to-metal ratio.”

Page 14, Lines 1-3

“The recombination process between Au NCs and Au(I)-thiolate complexes well explains the phenomenon observed in previous studies: [Au₂₂SR₁₈]⁰ and [Au₂₄SR₂₀]⁰ were successfully synthesized in an oxidative environment.”

Page 14, Lines 6-9

“It is also worth noting that another kind of recombination process was observed during the ligand-exchange-induced size/structure transformation process³⁵⁻³⁷, usually by the fusion of the metallic cores of two separate clusters, which is different from our observations in the etching reaction.”

Reference 35-37

35. Li, M., Tian, S., Wu, Z. & Jin, R. Peeling the core-shell Au₂₅ Nanocluster by reverse ligand-exchange. *Chem. Mater.* **28**, 1022 (2016).

36. Zeng, C., Chen, Y., Das, A. & Jin, R. Transformation chemistry of gold nanoclusters: from one

stable size to another. *J. Phys. Chem. Lett.* **6**, 2976 (2015).

37. Bootharaju, M. S., Joshi, C. P., Alhilaly, M. J. & Bakr, O. M. Switching a nanocluster core from hollow to nonhollow. *Chem. Mater.* **28**, 3292 (2016).

Comment 3. Can the authors comment on the generality of the elucidated mechanism? What factors play the most important role- temperature, specific ligands, etc?

Reply: Thank you very much for the insightful comments. We believe that the etching mechanism we elucidated in this paper can be widely applied to other reaction/ligand systems. It is applicable for previous observations in the etching process of metal nanoclusters (for example, the yield of fluorescent metal nanoclusters, such as Au₂₂SR₁₈ and Au₂₄SR₂₀, is higher in an oxidative environment (*J. Am. Chem. Soc.* **136**, 1246 (2014); *Angew. Chem. Int. Ed.* **55**, 11567 (2016)). In addition, reaction parameters should also be considered when designing future reaction systems. The reaction temperature will surely affect the reaction kinetics. As shown in Figure R4, at 50 °C, the etching process was greatly accelerated, where Au₂₅MHA₁₈ nanoclusters were completely decomposed within 4 hours. In addition to temperature, we want to particularly emphasize the choice of reaction pH in different ligand systems. As demonstrated in our paper, shorter Au(I)-SR complexes are crucial for the isoelectric addition reaction in Stage II to produce species with high ligand-to-metal ratios. This means that the Au(I)-SR complexes need to exist in the form of dissociated small molecules, rather than large precipitates in the reaction. The form of Au(I)-SR complexes in the water phase is highly dependent on the pH value of the solution. For example, for MHA, when the pH is less than 5.0, the Au(I)-MHA complexes will precipitate. When the ligand is glutathione (GSH), this pH can be as low as 2.0. This is why Au₂₂SG₁₈ can be produced at pH 2.5 (*J. Am. Chem. Soc.* **136**, 1246 (2014)).

Figure R4. ESI mass spectra of reaction solution at 50 °C.

Comment 4. References should be placed after the comma/period.

Reply: Thank you. The manuscript is prepared based on the *Nat. Commun.* format in which the reference is placed before the comma/period.

Comments by Reviewer #3:

General comments. *This publication reports an ESI- mass spectroscopic study of aqueous solutions of Au₂₅(MHA)₁₈ with excess added MHA at pH=9. In addition to the temporal development of high resolution mass spectra serially acquired over an unusually long time period ranging up to 700 hours after mixing, the authors report collision induced dissociation measurements and density functional theory calculations of selected species generated during the corresponding “etching” process. This leads sequentially to smaller clusters. Surprisingly there are also indications of addition reactions which set in at intermediate times.*

I have several problems with this manuscript which lead me to recommend rejection:

Reply: We are very grateful to the reviewer for his/her efforts in reviewing our manuscript and providing us with constructive comments and suggestions to further improve the quality of our paper.

Comment 1. *While pH is apparently held constant over more than 700 hours nothing is said about possibly changing solution concentration. What about solvent evaporation/condensation or adsorption of reactants on the container walls? What container material? To what extent is the kinetic dependence observed in Figure 2 (which by the way is almost unreadable) reproducible? How does it depend on temperature (was the solution thermostatted and held under constant humidity)?*

Reply: We fully agree that careful control of experimental conditions is crucial for data reproducibility. We used glass containers in our experiments. The container was saturated with air after the addition of the thiol ligands to initiate the etching process, and it was kept airtight. The solution for testing was drawn out by a syringe to avoid possible loss of solvent. The temperature was kept at 25 °C. The container was also kept in dark to avoid possible influence of light. More importantly, three batches of experiments were conducted to ensure the reproducibility of the etching process. We have included an explanation of condition control in the revised manuscript (in the part of Method), and we believe this can answer the queries on reproducibility. Thank you for this good suggestion.

We apologize for the figure issue, and hope that the reviewer can understand the difficulty of displaying all the 30 days' ESI-MS data in the figure. We have tried our best to make the figures as clear as possible. In addition, we have separated Figure 2a into 4 parts and shown them in Figure R5-R8 for your reference. The four separated figures are also included in our revised Supplementary Information for a better overview of the reaction process.

Figure R5. Enlarged view of ESI mass spectra in the range of 400-1000 m/z from 0 h to 48 h.

Figure R6. Enlarged view of ESI mass spectra in the range of 400-1000 m/z from 72 h to 720 h.

Figure R7. Enlarged view of ESI mass spectra in the range of 1000-3000 m/z from 0 h to 48 h.

Figure R8. Enlarged view of ESI mass spectra in the range of 1000-3000 m/z from 72 h to 720 h.

Revisions

Page 7, Lines 2-4

“Enlarged view of different parts are also shown in Supplementary Fig. 2-5 for a better overview of the ESI mass spectra in 30 days.”

Page 17, Lines 4-9

“The glass containers were used in the experiments. The container was saturated with air after adding the thiol ligands to initiate the etching process, and it was kept airtight. The solution for testing was drawn out by a syringe to avoid possible loss of solvent. The temperature was kept at 25 °C. The container was also kept in dark to avoid possible influence of light. Three batches of experiments were conducted to ensure the reproducibility of the etching process.”

Supplementary Fig. 2-5

Figure R5-R8 were included as Supplementary Fig. 2-5 in the revised Supplementary Information.

Comment 2. *MHA complexes were studied by negative ion mass spectroscopy relying on deprotonation of the MHA ligands for ionization. For example, primarily the tetraanionic species $[Au_{25}(MHA)_{18} - 4H^+]^{4-}$ was in fact initially probed - rather than “ $Au_{25}(MHA)_{18}$ ” or “ $Au_{25}(MHA)_{18}^-$ ” as discussed in the main text. This tetraanion is then etched to smaller trianions and dianions rather than to the neutral or monoanionic species discussed in the main text. Presumably the authors have chosen to neglect this in the main text to make things clearer for the reader. The opposite is in fact the case requiring this reader to go back and forth between supplement and main text to understand. Apart from terminology there is another possibly severer buried issue here: *J.Phys.Chem. A* 2020, 124, 5840–5848 has suggested that there are “charge isomers” in related MBA terminated Au₂₅ clusters, i.e. the cores of nominally equi m/z species can have a range of different charges. The authors should also report changes to other charge states bearing the same gold/ligand compositions (e.g. the parent tetraanion also comes as a (weaker) trianion and pentaanion).*

Reply: Thank you for these constructive comments/suggestions. It should be noted that the ionization states of the species captured by ESI-MS are different from that in the solution phase. Based on the mechanism of ESI-MS, the species in the solution phase are ionized through the desolvation process, which will change their surface charge states (*Angew. Chem. Int. Ed.* **58**, 11967 (2019)). One obvious evidence is that a species can show more than one signals with different charge states in the ESI mass spectrum. For example, in Figure R9, the species with the molecular formula $[Au_{25}(MHA)_{18}]^0$ show signals of $[Au_{25}MHA_{18}^0 - 5H^+]^{5-}$, $[Au_{25}MHA_{18}^0 - 4H^+]^{4-}$ and $[Au_{25}MHA_{18}^0 - 3H^+]^{3-}$ by losing different number of protons from the MHA ligands in the same ionization process. As a result, the species captured in the ESI mass spectra can only provide information of their molecular formulae, but not the corresponding ions in the solution phase. At present, to the best of our knowledge, there is no such method to precisely determine the ionization of metal nanoclusters capped by multi-carboxylic groups in the solution phase. Therefore, we can only discuss the evolution of species based on the molecular formula from the ESI mass spectra.

We agree with the reviewer that “charge isomers” are important issues that should be considered in molecular formula assignment. This is a good suggestion, and we have noticed this issue. When

simulating the isotopic pattern of $\text{Au}_{25}(\text{MHA})_{18}$ species, compared with $[\text{Au}_{25}(\text{MHA})_{18}]^-$, the mass of $[\text{Au}_{25}(\text{MHA})_{18}]^0$ differs by 1 Da. Since our experimental isotopic pattern best matches $[\text{Au}_{25}(\text{MHA})_{18}]^0$ instead of $[\text{Au}_{25}(\text{MHA})_{18}]^-$, we assign the starting nanocluster species as $[\text{Au}_{25}(\text{MHA})_{18}]^0$. As for the other observed species, we believe that the experimental signals can be perfectly fitted by the simulated isotopic pattern without such shifting (Supplementary Figure 7-23), indicating that the “charge isomers” can be ignored in these species.

We have included the related paper in the revised version as a reference.

Figure R9. Different degree of ionization in the same ESI-MS spectrum of $[\text{Au}_{25}(\text{MHA})_{18}]^0$.

Revisions:

Reference 19

19. Comby-Zerbino, C., Bertorelle, F., Dugourd, P., Antoine, R. & Chirot F. Structure and charge heterogeneity in isomeric $\text{Au}_{25}(\text{MBA})_{18}$ nanoclusters-insights from ion mobility and mass spectrometry. *J. Phys. Chem. A* **124**, 5840 (2020).

Comment 3. *I appreciate that this is probably one of the first studies in this field to observe long-term etching phenomena by mass spectrometry. But the authors devote a lot of additional space to making mechanistic statements which in my view are not sufficiently supported by the data. If they want to discuss mechanism they should provide a quantitative kinetic model and compare it to their observations.*

Reply: Thank you for your recognition of the novelty of our work. We agree with the reviewer that a quantitative kinetic model will greatly facilitate the understanding of the etching process. However, etching $\text{Au}_{25}(\text{MHA})_{18}$ can produce 10 species as final products in a 30-day reaction, and generate more species as intermediates. Moreover, the conversion from one species to another may involve several elementary reactions. Since it is a complicated reaction system involving the interconversion of multiple species, we are afraid that it will be very difficult (or almost impossible) to construct a quantitative model to explain all the observations presented in this study.

We propose to separate the entire reaction system into elementary reactions, as shown in Supplementary Scheme 1-3. It will be interesting to purify/separate the corresponding species by polyacrylamide gel electrophoresis (PAGE) and study their reaction kinetics in the elementary reactions. In this way, it will be possible to establish a quantitative kinetic model for each elementary reaction. This should be an interesting research topic in our future study, but we hope that the reviewer can understand the difficulty of accomplishing all these tasks in a single work and consider this as an out-of-scope topic of the current study.

In addition, our observations can provide some preliminary insights into the reaction kinetics. For example, we can use the data in Figure 4 to compare the relative lifetime of the species. There is an obvious trend in the lifetime of species with the same valence electron count, that is, the lifetime of larger isoelectric species is longer. This indicates that larger isoelectric species are more stable in (i.e., more resistant to) an oxidative environment, and explains why the recombination process by isoelectric addition is preferred in Stage II. We also conducted DFT calculation and control experiments to support our proposed recombination process (Figure 5). All our discussions on the reaction mechanisms are based on our experimental observations, which we believe are reasonable and self-consistent. Thank you very much for your insightful suggestions.

Reviewers' Comments:

Reviewer #1:

Remarks to the Author:

The revision has addressed my concern properly. The manuscript is ready for publication.

Reviewer #2:

Remarks to the Author:

The authors have addressed all reviewer comments appropriately and hence, I recommend the publication of this manuscript.

Reviewer #3:

None